# Accelerating Guided Diffusion Sampling with Splitting Numerical Methods

**Suttisak Wizadwongsa, Supasorn Suwajanakorn**
VISTEC, Thailand
{suttisak.w_s19, supasorn.s}@vistec.ac.th

## Abstract

*Guided diffusion* is a technique for conditioning the output of a diffusion model at sampling time without retraining the network for each specific task. However, one drawback of diffusion models, whether they are guided or unguided, is their slow sampling process. Recent techniques can accelerate unguided sampling by applying high-order numerical methods to the sampling process when viewed as differential equations. On the contrary, we discover that the same techniques do not work for guided sampling, and little has been explored about its acceleration. This paper explores the culprit of this problem and provides a solution based on operator splitting methods, motivated by our key finding that classical high-order numerical methods are unsuitable for the conditional function. Our proposed method can re-utilize the high-order methods for guided sampling and can generate images with the same quality as a 250-step DDIM baseline using 32-58% less sampling time on ImageNet256. We also demonstrate usage on a wide variety of conditional generation tasks, such as text-to-image generation, colorization, inpainting, and super-resolution.

## 1 Introduction

A family of generative models known as diffusion models has recently gained a lot of attention with state-of-the-art image generation quality (Dhariwal & Nichol, 2021). *Guided diffusion* is an approach for controlling the output of a trained diffusion model for conditional generation tasks without retraining its network. By engineering a task-specific conditional function and modifying only the sampling procedure, guided diffusion models can be used in a variety of applications, such as class-conditional image generation (Dhariwal & Nichol, 2021; Kawar et al., 2022), text-to-image generation (Nichol et al., 2022), image-to-image translation (Zhao et al., 2022), inpainting (Chung et al., 2022a), colorization (Song et al., 2020b), image composition (Sasaki et al., 2021), adversarial purification (Wang et al., 2022; Wu et al., 2022) and super-resolution (Choi et al., 2021).

One common drawback of both guided and regular "unguided" diffusion models is their slow sampling processes, usually requiring hundreds of iterations to produce a single image. Recent speed-up attempts include improving the noise schedule (Nichol & Dhariwal, 2021; Watson et al., 2021), redefining the diffusion process to be non-Markovian, thereby allowing a deterministic sampling process Song et al. (2020a), network distillation that teaches a student model to simulate multiple sampling steps of a teacher model Salimans & Ho (2022); Luhman & Luhman (2021), among others. Song et al. (2020a) show how each sampling step can be expressed as a first-order numerical step of an ordinary differential equation (ODE). Similarly, Song et al. (2020b) express the sampling of a score-based model as solving a stochastic differential equation (SDE). By regarding the sampling process as an ODE/SDE, many high-order numerical methods have been suggested, such as Liu et al. (2022), Zhang & Chen (2022), and Zhang et al. (2022) with impressive results on unguided diffusion models. However, when applied to guided diffusion models, these methods produce surprisingly poor results (see Figure 1)—given a few number of steps, those high-order numerical methods actually perform worse than low-order methods.

Guided sampling differs from the unguided one by the addition of the gradients of the conditional function to its sampling equation. The observed performance decline thus suggests that classical high-order methods may not be suitable for the conditional function and, consequently, the guided

Number
of steps

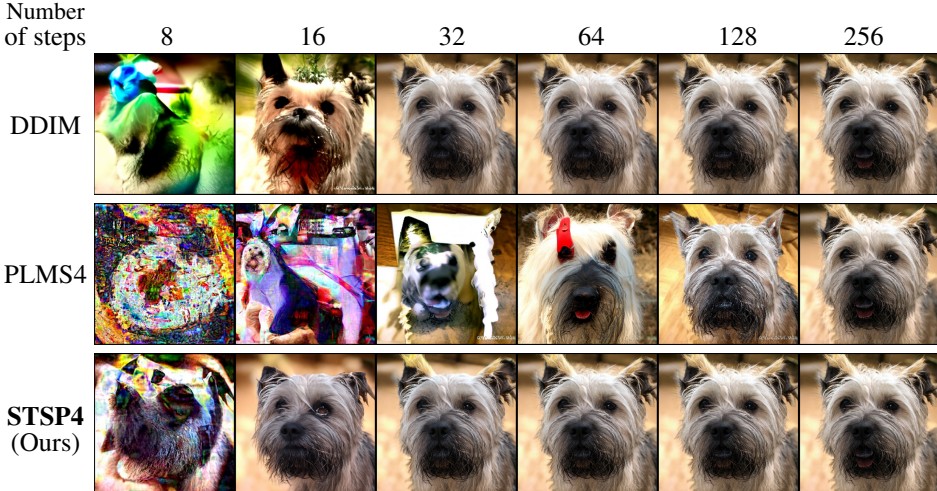

Figure 1: Generated samples of a classifier-guided diffusion model trained on ImageNet256 using 8-256 sampling steps from different sampling methods. Our technique, STSP4, produces high-quality results in a fewer number of steps.

sampling equation as a whole. Our paper tests this hypothesis and presents an approach to accelerating guided diffusion sampling. The key idea is to use an *operator splitting* method to split the less well-behaved conditional function term from the standard diffusion term and solve them separately. This approach not only allows re-utilizing the successful high-order methods on the diffusion term but also provides us with options to combine different specialized methods for each term to maximize performance. Note that splitting methods have also been explored by Dockhorn et al. (2022) to solve unguided diffusion SDEs, but our work focuses on accelerating *guided* diffusion ODEs.

Our design process includes comparing different splitting methods and numerical methods for each split term. When tested on ImageNet, our approach achieves the same level of image quality as a DDIM baseline while reducing the sampling time by approximately 32-58%. Compared with other sampling methods using the same sampling time, our approach provides better image quality as measured by LPIPS, FID, and Perception/Recall. With only minimal modifications to the sampling equation, we also show successful acceleration on various conditional generation tasks.

## 2 BACKGROUND

This section provides a high-level summary of the theoretical foundation of diffusion models as well as numerical methods that have been used for diffusion models. Here we briefly explain a few that contribute to our method.

### 2.1 DIFFUSION MODELS

Assuming that $x_0$ is a random variable from the data distribution we wish to reproduce, diffusion models define a sequence of Gaussian noise degradation of $x_0$ as random variables $x_1, x_2, ..., x_T$, where $x_t \sim \mathcal{N}(\sqrt{1 - \beta_t}x_{t-1}, \beta_t\mathbf{I})$ and $\beta_t \in [0, 1]$ are parameters that control the noise levels. With a property of Gaussian distribution, we can express $x_t$ directly as a function of $x_0$ and noise $\epsilon \sim \mathcal{N}(0, \mathbf{I})$ by $x_t = \sqrt{\bar{\alpha}_t}x_0 + \sqrt{1 - \bar{\alpha}_t}\epsilon$, where $\bar{\alpha}_t = \prod_{i=1}^{t}(1 - \beta_i)$. By picking a sufficiently large $T$ (e.g., 1,000) and an appropriate set of $\beta_t$, we can assume $x_T$ is a standard Gaussian distribution. The main idea of diffusion model generation is to sample a Gaussian noise $x_T$ and use it to reversely sample $x_{T-1}, x_{T-2}, ...$ until we obtain $x_0$, which belongs to our data distribution.

Ho et al. (2020) propose Denoising Diffusion Probabilistic Model (DDPM) and explain how to employ a neural network $\epsilon_\theta(x_t, t)$ to predict the noise $\epsilon$ that is used to compute $x_t$. To train the network, we sample a training image $x_0$, $t$, and $\epsilon$ to compute $x_t$ using the above relationship. Then, we optimize our network $\epsilon_\theta$ to minimize the difference between the predicted and real noise, i.e., $\|\epsilon - \epsilon_\theta(x_t, t)\|^2$.

Song et al. (2020a) introduce Denoising Diffusion Implicit Model (DDIM), which uses the network $\epsilon_\theta$ to deterministically obtain $x_{t-1}$ given $x_t$. The DDIM generative process can be written as

$$x_{t-1} = \sqrt{\frac{\bar{\alpha}_{t-1}}{\bar{\alpha}_t}} \left(x_t - \sqrt{1 - \bar{\alpha}_t}\epsilon_\theta(x_t, t)\right) + \sqrt{1 - \bar{\alpha}_{t-1}}\epsilon_\theta(x_t, t). \tag{1}$$

This formulation could be used to skip many sampling steps and boost sampling speed. To turn this into an ODE, we rewrite Equation 1 as:

$$\frac{x_{t-\Delta t}}{\sqrt{\bar{\alpha}_{t-\Delta t}}} = \frac{x_t}{\sqrt{\bar{\alpha}_t}} + \left(\sqrt{\frac{1 - \bar{\alpha}_{t-\Delta t}}{\bar{\alpha}_{t-\Delta t}}} - \sqrt{\frac{1 - \bar{\alpha}_t}{\bar{\alpha}_t}}\right)\epsilon_\theta(x_t, t), \tag{2}$$

which is now equivalent to a numerical step in solving an ODE. To derive the corresponding ODE, we can re-parameterize $\sigma_t = \sqrt{1 - \bar{\alpha}_t}/\sqrt{\bar{\alpha}_t}$, $\bar{x}(t) = x_t/\sqrt{\bar{\alpha}_t}$ and $\bar{\epsilon}_\sigma(\bar{x}) = \epsilon_\theta(x_t, t)$, yielding $\bar{x}(t - \Delta t) - \bar{x}(t) = (\sigma_{t-\Delta t} - \sigma_t)\bar{\epsilon}_\sigma(\bar{x})$. By letting $(\sigma_{t-\Delta t} - \sigma_t) \to 0$, the ODE becomes:

$$\frac{d\bar{x}}{d\sigma} = \bar{\epsilon}_\sigma(\bar{x}). \tag{3}$$

Note that this change of variables is equivalent to an exponential integrator technique described in both Zhang & Chen (2022) and Lu et al. (2022). Since $x_t$ and $\bar{x}(t)$ have the same value at $t = 0$, our work can focus on solving $\bar{x}(t)$ rather than $x_t$. Many numerical methods can be applied to the ODE Equation 3 to accelerate diffusion sampling. We next discuss some of them that are relevant.

## 2.2 NUMERICAL METHODS

**Euler's Method** is the most basic numerical method. A forward Euler step is given by $\bar{x}_{n+1} = \bar{x}_n + \Delta\sigma\bar{\epsilon}_\sigma(\bar{x}_n)$. When the forward Euler step is applied to the ODE Equation 3, we obtain the DDIM formulation (Song et al., 2020a).

**Heun's Method**, also known as the trapezoid rule or improved Euler, is given by: $\bar{x}_{n+1} = \bar{x}_n + \frac{\Delta\sigma}{2}(e_1 + e_2)$, where $e_1 = \bar{\epsilon}_\sigma(\bar{x}_n)$ and $e_2 = \bar{\epsilon}_\sigma(\bar{x}_n + \Delta\sigma e_1)$. This method splits Euler's method into two steps to improve accuracy. Many papers have used this method on diffusion models, including Algorithm 1 in Karras et al. (2022) and DPM-Solver-2 in Lu et al. (2022). This method is also the simplest case of Predictor-Corrector methods used in Song et al. (2020b).

**Runge-Kutta Methods** represent a class of numerical methods that integrate information from multiple hidden steps and provide high accuracy results. Heun's method also belongs to a family of $2^{nd}$-order Runge-Kutta methods (RK2). The most well-known variant is the $4^{th}$-order Runge-Kutta method (RK4), which is written as follows:

$$e_1 = \bar{\epsilon}_\sigma(\bar{x}_n), \quad e_2 = \bar{\epsilon}_\sigma\left(\bar{x}_n + \frac{\Delta\sigma}{2}e_1\right), \quad e_3 = \bar{\epsilon}_\sigma\left(\bar{x}_n + \frac{\Delta\sigma}{2}e_2\right), \quad e_4 = \bar{\epsilon}_\sigma\left(\bar{x}_n + \Delta\sigma e_3\right),$$

$$\bar{x}_{n+1} = \bar{x}_n + \frac{\Delta\sigma}{6}(e_1 + 2e_2 + 2e_3 + e_4). \tag{4}$$

This method has been tested on diffusion models in Liu et al. (2022) and Salimans & Ho (2022), but it has not been used as the main proposed method in any paper.

**Linear Multi-Step Method**, similar to the Runge-Kutta methods, aims to combine information from several steps. However, rather than evaluating new hidden steps, this method uses the previous steps to estimate the new step. The $1^{st}$-order formulation is the same as Euler's method. The $2^{nd}$-order formulation is given by

$$\bar{x}_{n+1} = \bar{x}_n + \frac{\Delta\sigma}{2}\left(3e_0 - e_1\right), \tag{5}$$

while the $4^{th}$-order formulation is given by

$$\bar{x}_{n+1} = \bar{x}_n + \frac{\Delta\sigma}{24}(55e_0 - 59e_1 + 37e_2 - 9e_3), \tag{6}$$

where $e_k = \bar{\epsilon}_\sigma(\bar{x}_{n-k})$. These formulations are designed for a constant $\Delta\sigma$ in each step. However, our experiments and previous work that uses this method (e.g., Liu et al. (2022); Zhang & Chen

(2022)) still show good results when this assumption is not strictly satisfied, i.e., when $\Delta\sigma$ is not constant. We will refer to these formulations as PLMS (Pseudo Linear Multi-Step) for the rest of the paper, like in Liu et al. (2022). A similar linear multi-step method for non-constant $\Delta\sigma$ can also be derived using a technique used in Zhang & Chen (2022), which we detail in Appendix B. This non-constant version can improve upon PLMS slightly, but it is not as flexible because we have to re-derive the update rule every time the $\sigma$ schedule changes.

# 3   SPLITTING METHODS FOR GUIDED DIFFUSION MODELS

This section introduces our technique that uses splitting numerical methods to accelerate guided diffusion sampling. We first focus our investigation on *classifier-guided* diffusion models for class-conditional generation and later demonstrate how this technique can be used for other conditional generation tasks in Section 4.3. Like any guided diffusion models, classifier-guided models (Dhariwal & Nichol, 2021) share the same training objective with regular unguided models with no modifications to the training procedure; but the sampling process is guided by an additional gradient signal from an external classifier to generate class-specific output images. Specifically, the sampling process is given by

$$\hat{\epsilon} = \epsilon_\theta(x_t) - \sqrt{1 - \bar{\alpha}_t}\nabla_x \log p_\phi(c|x_t), \quad x_{t-1} = \sqrt{\bar{\alpha}_{t-1}}\left(\frac{x_t - \sqrt{1 - \bar{\alpha}_t}\hat{\epsilon}}{\sqrt{\bar{\alpha}_t}}\right) + \sqrt{1 - \bar{\alpha}_{t-1}}\hat{\epsilon}, \quad (7)$$

where $p_\phi(c|x_t)$ is a classifier model trained to output the probability of $x_t$ belonging to class $c$. As discussed in the previous section, we can rewrite this formulation as a "guided ODE":

$$\frac{d\bar{x}}{d\sigma} = \bar{\epsilon}_\sigma(\bar{x}) - \nabla f_\sigma(\bar{x}), \quad (8)$$

where $f_\sigma(\bar{x}) = \frac{\sigma}{\sqrt{\sigma^2+1}}\log p_\phi(c|x_t)$. We refer to $f_\sigma$ as the conditional function, which can be substituted with other functions for different tasks. After obtaining the ODE form, any numerical solver mentioned earlier can be readily applied to accelerate the sampling process. However, we observe that classical high-order numerical methods (e.g., PLMS4, RK4) fail to accelerate this task (see Figure 1) and even perform worse than the baseline DDIM.

We hypothesize that the two terms in the guided ODE may have different numerical behaviors with the conditional term being less suitable to classical high-order methods. We speculate that the difference could be partly attributed to how they are computed: $\nabla f_\sigma(\bar{x})$ is computed through back-propagation, whereas $\bar{\epsilon}_\sigma(\bar{x})$ is computed directly by evaluating a network. One possible solution to handle terms with different behaviors is the so-called operator splitting method, which divides the problem into two subproblems:

$$\frac{dy}{d\sigma} = \bar{\epsilon}_\sigma(y), \quad \frac{dz}{d\sigma} = -\nabla f_\sigma(z). \quad (9)$$

We call these the *diffusion* and *condition* subproblems, respectively. This method allows separating the hard-to-approximate $\nabla f_\sigma(z)$ from $\bar{\epsilon}_\sigma(y)$ and solving them separately in each time step. Importantly, this helps reintroduce the effective use of high-order methods on the diffusion subproblem as well as provides us with options to combine different specialized methods to maximize performance. We explore two most famous first- and second-order splitting techniques for our task:

## 3.1   LIE-TROTTER SPLITTING (LTSP)

Our first example is the simple first-order Lie-Trotter splitting method (Trotter, 1959), which expresses the splitting as

$$\frac{dy}{d\sigma} = \bar{\epsilon}_\sigma(y), \qquad\qquad y(\sigma_n) = \bar{x}_n, \qquad\qquad \sigma \in [\sigma_{n+1}, \sigma_n] \qquad (10)$$

$$\frac{dz}{d\sigma} = -\nabla f_\sigma(z), \qquad\qquad z(\sigma_n) = y(\sigma_{n+1}), \qquad\qquad \sigma \in [\sigma_{n+1}, \sigma_n] \qquad (11)$$

with the solution of this step being $\bar{x}_{n+1} = z(\sigma_{n+1})$. Note that $\sigma_n$ is a decreasing sequence. Here Equation 10 is the same as Equation 3, which can be solved using any high-order numerical method,

| **Algorithm 1:** Lie-Trotter Splitting (LTSP) |
| --- |
| sample $\bar{x}_0 \sim \mathcal{N}(0, \sigma_{\max}^2 \mathbf{I})$ ; |
| **for** $n \in \{0, ..., N-1\}$ **do** |
|    $y_{n+1} = \text{PLMS}(\bar{x}_n, \sigma_n, \sigma_{n+1}, \bar{\epsilon}_\sigma)$; |
|    $\bar{x}_{n+1} = y_{n+1} - (\sigma_{n+1} - \sigma_n)\nabla f(y_{n+1})$ ; |
| **end** |
| **Result:** $\bar{x}_N$ |

| **Algorithm 2:** Strang Splitting (STSP) |
| --- |
| sample $\bar{x}_0 \sim \mathcal{N}(0, \sigma_{\max}^2 \mathbf{I})$ ; |
| **for** $n \in \{0, ..., N-1\}$ **do** |
|    $z_{n+1} = \bar{x}_n - \frac{(\sigma_{n+1} - \sigma_n)}{2}\nabla f(\bar{x}_n)$ ; |
|    $y_{n+1} = \text{PLMS}(z_{n+1}, \sigma_n, \sigma_{n+1}, \bar{\epsilon}_\sigma)$; |
|    $\bar{x}_{n+1} = y_{n+1} - \frac{(\sigma_{n+1} - \sigma_n)}{2}\nabla f(y_{n+1})$ ; |
| **end** |
| **Result:** $\bar{x}_N$ |

e.g., PLMS. For Equation 11, we can use a forward Euler step:

$$z_{n+1} = z_n - \Delta\sigma\nabla f_\sigma(z_n). \tag{12}$$

This is equivalent to a single iteration of standard gradient descent with a learning rate $\Delta\sigma$. This splitting scheme is summarized by Algorithm 1. We investigate different numerical methods for each subproblem in Section 4.1.

## 3.2 STRANG SPLITTING (STSP)

Strang splitting (or Strang-Marchuk) (Strang, 1968) is one of the most famous and widely used operator splitting methods. This second-order splitting works as follows:

$$\frac{dz}{d\sigma} = -\nabla f_\sigma(z), \qquad\qquad z(\sigma_n) = \bar{x}_n, \qquad \sigma \in \left[\frac{1}{2}(\sigma_n + \sigma_{n+1}), \sigma_n\right] \tag{13}$$

$$\frac{dy}{d\sigma} = \bar{\epsilon}_\sigma(y), \qquad y(\sigma_n) = z\left(\frac{1}{2}(\sigma_n + \sigma_{n+1})\right), \qquad\qquad \sigma \in [\sigma_{n+1}, \sigma_n] \tag{14}$$

$$\frac{d\tilde{z}}{d\sigma} = -\nabla f_\sigma(\tilde{z}), \quad \tilde{z}\left(\frac{1}{2}(\sigma_n + \sigma_{n+1})\right) = y(\sigma_{n+1}), \qquad \sigma \in \left[\sigma_{n+1}, \frac{1}{2}(\sigma_n + \sigma_{n+1})\right] \tag{15}$$

Instead of solving each subproblem for a full step length, we solve the condition subproblem for half a step before and after solving the diffusion subproblem for a full step. In theory, we can swap the order of operations without affecting convergence, but it is practically cheaper to compute the condition term twice rather than the diffusion term twice because $f_\sigma$ is typically a smaller network compared to $\bar{\epsilon}_\sigma$. The Strange splitting algorithm is shown in Algorithm 2. This method can be shown to have better accuracy than the Lie-Trotter method, as proven in Appendix N. Although it requires evaluating the condition term twice per step in exchange for improved image quality. We assess this trade-off in the experiment section.

## 4 EXPERIMENTS

Extending on our observation that classical high-order methods failed on guided sampling, we conducted a series of experiments to investigate this problem and evaluate our solution. Section 4.1 uses a simple splitting method (first-order LTSP) to study the effects that high-order methods have on each subproblem, leading to our key finding that *only* the conditional subproblem is less suited to classical high-order methods. This section also determines the best combination of numerical methods for the two subproblems under LTSP splitting. Section 4.2 explores improvements from using a higher-order splitting method and compares our best scheme to previous work. Finally, Section 4.3 applies our approach to a variety of conditional generation tasks with minimal changes.

For our comparison, we use pre-trained state-of-the-art diffusion models and classifiers from Dhariwal & Nichol (2021), which were trained on the ImageNet dataset (Russakovsky et al., 2015) with 1,000 total sampling steps. We treat full-path samples from a classifier-guided DDIM at 1,000 steps as reference solutions. Then, the performance of each configuration is measured by the image similarity between its generated samples using fewer steps and the reference DDIM samples, both starting from the same initial noise map. Given the same sampling time, we expect configurations with better performance to better match the full DDIM. We measure image similarity using Learned Perceptual Image Patch Similarity (LPIPS) (Zhang et al., 2018) (lower is better) and measure sampling time on a single NVIDIA RTX 3090 and a 24-core AMD Threadripper 3960x.

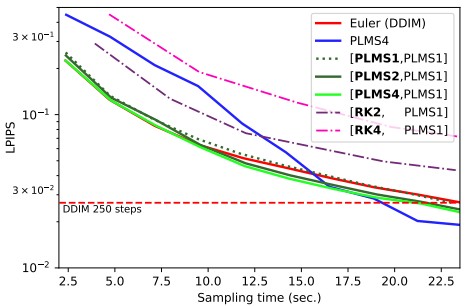 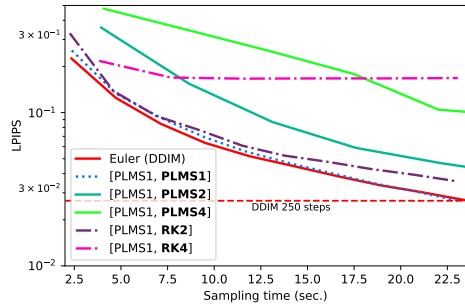

(a) Varying the method for the diffusion subproblem   (b) Varying the method for the condition subproblem

Figure 2: Comparison of different combinations of numerical methods under LTSP splitting for guided diffusion sampling. We plot LPIPS against the sampling time. [A, B] denotes the use of method A in the diffusion subproblem and method B in the condition subproblem. The red dotted lines indicate a reference DDIM score obtained from 250 sampling steps, which produce images visually close to those from 1,000 steps.

## 4.1 FINDING A SUITABLE NUMERICAL METHOD FOR EACH SUBPROBLEM

To study the effects of different numerical methods on each subproblem of the guided ODE (Equation 8), we use the simplest Lie-Trotter splitting, which itself requires no additional network evaluations. This controlled experiment has two setups: a) we fix the numerical method for the condition subproblem (Equation 11) to first-order PLMS1 (Euler's method) and vary the numerical method for the diffusion subproblem (Equation 10), and conversely b) we fix the method for the diffusion subproblem and vary the method for the condition subproblem. The numerical method options are Euler's method (PLMS1), Heun's method (RK2), 4th order Runge-Kutta's method (RK4), and 2nd/4th order pseudo linear multi-step (PLMS2/PLMS4). We report LPIPS vs. sampling time of various numerical combinations on a diffusion model trained on ImageNet 256×256 in Figure 2. The red dotted lines indicate a reference DDIM score obtained from 250 sampling steps, a common choice that produces good samples that are perceptually close to those from a full 1,000-step DDIM (Dhariwal & Nichol, 2021; Nichol & Dhariwal, 2021).

Given a long sampling time, non-split PLMS4 performs better than the DDIM baseline. However, when the sampling time is reduced, the image quality of PLMS4 rapidly decreases and becomes much worse than that of DDIM, especially under 15 seconds in Figure 2. When we split the ODE and solve both subproblems using first-order PLMS1 (Euler), the performance is close to that of DDIM, which is also considered first-order but without any splitting. This helps verify that merely splitting the ODE does not significantly alter the sampling speed.

In the setup a), when RK2 and RK4 are used for the diffusion subproblem, they also perform worse than the DDIM baseline. This slowdown is caused by the additional evaluations of the network by these methods, which outweigh the improvement gained in each longer diffusion step. Note that if we instead measure the image quality with respect to the number of diffusion steps, RK2 and RK4 can outperform other methods (Appendix E); however, this is not our metric of interest. On the other hand, PLMS2 and PLMS4, which require no additional network evaluations, are about 8-10% faster than DDIM and can achieve the same LPIPS score as the DDIM that uses 250 sampling steps in 20-26 fewer steps. Importantly, when the sampling time is reduced, their performance does not degrade rapidly like the non-split PLMS4 and remains at the same level as DDIM.

In the setup b) where we vary the numerical method for the condition subproblem, the result reveals an interesting contrast—none of the methods beats DDIM and some even make the sampling diverged [PLMS1, RK4]. These findings suggest that the gradients of conditional functions are less "compatible" with classical high-order methods, especially when used with a small number of steps. This phenomenon may be related to the "stiffness" condition of ODEs, which we discuss further in Section 5. For the remainder of our experiments, we will use the combination [PLMS4, PLMS1] for the diffusion and condition subproblems, respectively.

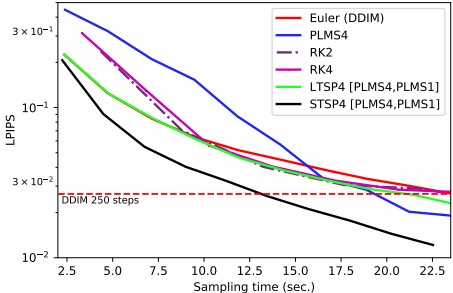

Figure 3: Comparison of different numerical methods for guided diffusion sampling.

| | Sampling time within | | | |
|---|---|---|---|---|
| | 5 sec. | 10 sec. | 15 sec. | 20 sec. |
| DDIM | 0.116 | 0.062 | 0.043 | 0.033 |
| PLMS4 | 0.278 | 0.141 | 0.057 | 0.026 |
| RK2 | 0.193 | 0.059 | 0.036 | 0.028 |
| RK4 | 0.216 | 0.054 | 0.039 | 0.028 |
| **LTSP4** | 0.121 | 0.058 | 0.037 | 0.028 |
| **STSP4** | **0.079** | **0.035** | **0.022** | **0.013** |

Table 1: Average LPIPS when the sampling time is limited to be under 5 - 20 seconds.

## 4.2 IMPROVED SPLITTING METHOD

This experiment investigates improvements from using a high-order *splitting* method, specifically the Strang splitting method, with the numerical combination [PLMS4, PLMS1] and compares our methods to previous work. Note that besides DDIM Dhariwal & Nichol (2021), no previous work is specifically designed for accelerating *guided* sampling, thus the baselines in this comparison are only adaptations of the core numerical methods used in those papers. And to our knowledge, no prior guided-diffusion work uses splitting numerical methods. Non-split numerical method baselines are PLMS4, which is used in Liu et al. (2022), RK2, which is used in Karras et al. (2022); Lu et al. (2022), and higher-order RK4. We report the LPIPS scores of these methods with respect to the sampling time in Figure 3 and Table 1.

Without any splitting, PLMS4, RK2 and RK4 show significantly poorer image quality when used with short sampling times $< 10$ seconds. The best performer is our Strang splitting (STSP4), which can reach the same quality as 250-step DDIM while using 32-58% less sampling time. STSP4 also obtains the highest LPIPS scores for sample times of 5, 10, 15, and 20 seconds. More statistical details and comparison with other split combinations are in Appendix F, G.

In addition, we perform a quantitative evaluation for class-conditional generation by sampling 50,000 images based on uniformly chosen class conditions with a small number of sampling steps and evaluating the Fenchel Inception Distance (FID) Heusel et al. (2017) (lower is better) and the improved precision/recall Kynkäänniemi et al. (2019) (higher is better) against the ImageNet test set at 128, 256, and 512 resolutions. Following (Dhariwal & Nichol, 2021), we use a 25-step DDIM as a baseline, which already produces visually reasonable results. As PLMS and LTSP require the same number of network evaluations as the DDIM, they are used also with 25 steps. For STSP with a slower evaluation time, it is only allowed 20 steps, which is the highest number of steps such that its sampling time is within that of the baseline 25-step DDIM. Here LTSP2 and STSP2 are Lie-Trotter and Strang splitting methods with the combination [PLMS2, PLMS1]. In Table 2, we report the results for three different ImageNet resolutions and the average sampling time per image in seconds.

Our STSP4 performs best on all measurements except Recall on ImageNet512. On ImageNet512, PLMS4 has the highest Recall score but a poor FID of 16, indicating that the generated images have good distribution coverage but may poorly represent the real distribution. On ImageNet256, STSP4 can yield 4.49 FID in 20 steps, compared to 4.59 FID in 250 steps originally reported in the paper (Dhariwal & Nichol, 2021). Our STSP4 is about 9.4× faster when tested on the same machine.

## 4.3 SPLITTING METHODS IN OTHER TASKS

Besides class-conditional generation, our approach can also accelerate any conditional image generation as long as the gradient of the conditional function can be defined. We test our approach on four tasks: text-to-image generation, image inpainting, colorization, and super-resolution.

**Text-to-image generation:** We use a pre-trained text-to-image Disco-Diffusion (Letts et al., 2021) based on Crowson (2021), which substitutes the classifier output with the dot product of the image and caption encodings from CLIP (Radford et al., 2021). For more related experiments on Stable-Diffusion (Rombach et al., 2022), please refer to Appendix L, M.

| Method | Steps | Time | FID | Prec | Rec |
|---|---|---|---|---|---|
| **ImageNet128** | | | | | |
| DDIM | 25 | 0.55 | 6.69 | 0.78 | 0.49 |
| PLMS2 | 25 | 0.57 | 5.71 | 0.80 | 0.51 |
| PLMS4 | 25 | 0.57 | 4.97 | 0.80 | 0.53 |
| **LTSP2** | 25 | 0.55 | 5.14 | 0.81 | 0.51 |
| **LTSP4** | 25 | 0.55 | 3.85 | **0.81** | **0.54** |
| **STSP2** | 20 | 0.54 | 5.33 | 0.80 | 0.52 |
| **STSP4** | 20 | 0.54 | **3.78** | **0.81** | **0.54** |
| *ADM-G* | *250* | *5.59** | *2.97* | *0.78* | *0.59* |

| Method | Steps | Time | FID | Prec | Rec |
|---|---|---|---|---|---|
| **ImageNet256** | | | | | |
| DDIM | 25 | 1.99 | 5.47 | 0.80 | 0.47 |
| PLMS4 | 25 | 2.05 | 4.71 | 0.82 | 0.49 |
| **STSP4** | 20 | 1.95 | **4.49** | **0.83** | **0.50** |
| *ADM-G* | *250* | *20.9** | *4.59* | *0.82* | *0.50* |
| **ImageNet512** | | | | | |
| DDIM | 25 | 5.56 | 9.07 | 0.81 | 0.42 |
| PLMS4 | 25 | 5.78 | 16.00 | 0.75 | **0.51** |
| **STSP4** | 20 | 5.13 | **8.24** | **0.83** | 0.45 |
| *ADM-G* | *250* | *56.2** | *7.72* | *0.87* | *0.42* |

Table 2: Comparison of different numerical methods using a few steps on guided diffusion sampling. Our methods and the best scores are highlighted in bold. We provide the reported scores from Dhariwal & Nichol (2021) using 250 sampling steps, referred to as ADM-G in their paper. *ADM-G's sampling times are measured using our machine.

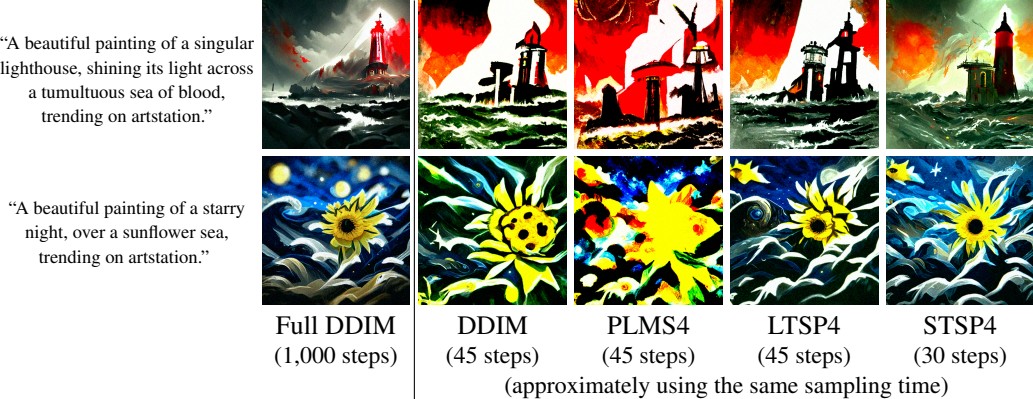

"A beautiful painting of a singular lighthouse, shining its light across a tumultuous sea of blood, trending on artstation."

"A beautiful painting of a starry night, over a sunflower sea, trending on artstation."

| Full DDIM (1,000 steps) | DDIM (45 steps) | PLMS4 (45 steps) | LTSP4 (45 steps) | STSP4 (30 steps) |
|---|---|---|---|---|

(approximately using the same sampling time)

Figure 4: Text-to-image generation using different sampling methods.

**Image inpainting & colorization:** For these two tasks, we follow the techniques proposed in Song et al. (2020b) and Chung et al. (2022a), which improves the conditional functions of both tasks with "manifold constraints." We use the same diffusion model Dhariwal & Nichol (2021) trained on ImageNet as our earlier Experiments 4.1, 4.2.

**Super-resolution:** We follow the formulation from ILVR (Choi et al., 2021) combined with the manifold contraints Chung et al. (2022a), and also use our earlier ImageNet diffusion model.

Figure 4 compares our techniques, LTSP4 and STSP4, with the DDIM baseline and PLMS4 on text-to-image generation. Each result is produced using a fixed sampling time of about 26 seconds. STSP4, which uses 30 diffusion steps compared to 45 in the other methods, produces more realistic results with color contrast that is more similar to the full DDIM references'. Figure 5 shows that our STSP4 produces more convincing results than the DDIM baseline with fewer artifacts on the other three tasks while using the same 5 second sampling time. Implementation details, quantitative evaluations, and more results are in Appendix J, K.

## 5 DISCUSSION

Our findings show that when the sampling ODE consists of multiple terms from different networks, their numerical behaviors can be different and treating them separately can be more optimal. Another promising direction is to improve the behavior of the gradient of the conditional function / classifier itself and study whether related properties such as adversarial robustness or gradient smoothness can induce the desirable temporal smoothness in the sampling ODE. However, it is not yet clear what specific characteristics of the behavior play an important role. This challenge may be related to a

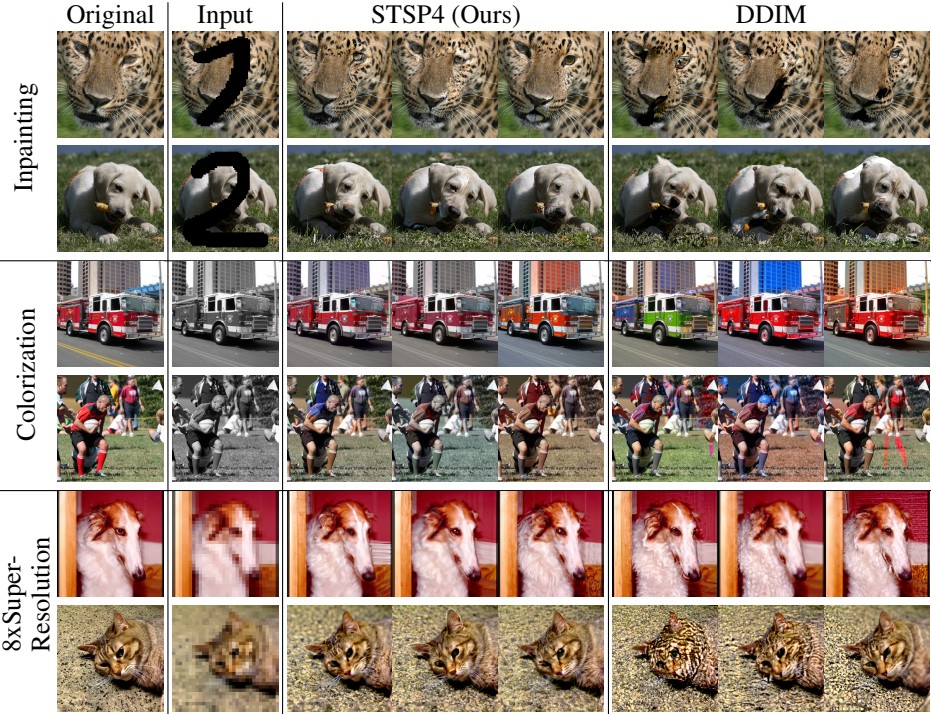

Figure 5: Guided-diffusion results of our STSP4 and DDIM on inpainting, colorization, and super-resolution. Both methods were limited to use approximately the same sampling time.

condition called "stiffness" in solving ODEs Ernst & Gerhard (2010), which lacks a clear definition but describes the situation where explicit numerical methods, such as RK or PLMS, require a very small step size *even in* regions with smooth curvature.

As an alternative to the classifier-guided model, Ho & Salimans (2021) propose a classifier-free model that can perform conditional generation without a classifier while remaining a generative model. This model can utilize high-order methods as no classifier is involved, but it requires evaluating the classifier-free network twice per step, which is typically more expensive than evaluating a normal diffusion model and a classifier. It is important to note that our accelerating technique and classifier-free models are *not* mutually exclusive, and one can still apply a conditional function and our splitting technique to guide a classifier-free model in a direction it has not been trained for.

While our paper only focuses on ODEs derived from the deterministic sampling of DDIM, one can convert SDE-based diffusion models to ODEs (Karras et al., 2022) and still use our technique. More broadly, we can accelerate any diffusion model that can be expressed as a differential equation with a summation of two terms. When these terms behave differently, the benefit from splitting can be substantial. Nevertheless, our findings are based on common, existing models and $\sigma$ schedule from Dhariwal & Nichol (2021). Further investigation into the impact of the $\sigma$ schedule or different types and architectures of diffusion models is still required.

## 6  CONCLUSION

In this paper, we investigate the failure to accelerate guided diffusion sampling of classical high-order numerical methods and propose a solution based on splitting numerical methods. We found that the gradients of conditional functions are less suitable to classical high-order numerical methods and design a technique based on Strang splitting and a combination of forth- and first-order numerical methods. Our method achieves better LPIPS and FID scores than previous work given the same sampling time and is 32-58% faster than a 250-step DDIM baseline. Our technique can successfully accelerate a variety of tasks, such as text-to-image generation, inpainting, colorization, and super-resolution.

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
