# OpenReview forum: "Accelerating Guided Diffusion Sampling with Splitting Numerical Methods"
_ICLR.cc/2023/Conference — ICLR 2023 poster_

### Official Review · Reviewer_PHsB · 2022-10-24

**Confidence:** 4
**Correctness:** 3
**Technical Novelty And Significance:** 2
**Empirical Novelty And Significance:** 2
**Recommendation:** 6

**Clarity, Quality, Novelty And Reproducibility:**

The paper is generally well-written. The demonstrated phenomenon of the paper remains mysterious. It would be helpful to elaborate on the behavior of the condition function. Intuitively, since we want to minimize the discretization error of an ODE, a higher order method could lead to better performance.

**Strength And Weaknesses:**

### Strong points

- The paper applies the operator splitting method to the classifier-guided generation. They study various combinations of solvers of the individual parts, and the best combination outperforms the baseline DDIM.

- The proposed method consistently improved over the baselines across image resolutions and tasks.



### Weak points

- The current dominating method is classifier-free generation, such as stable diffusion. The problem of classifier-guided diffusion models seems to have lower practical value. As the authors mention in Section 5, classifier-free involves more expensive function evaluation per step. However, they allow higher-order solvers for all components. Could the authors compare the FID/sampling time of the two methods, in order to demonstrate the utility of classifier-guided generation. (I understand it could be hard to complete the experiments in the short rebuttal time, but it really boosts the utility of classifier-guided generation if it has certain advantages.)

- The "stiffness" reasoning of the condition function is vague and insufficient. The phenomenon is the major point in the paper, and it would be very helpful to dive deeper into this problem: why does the condition function perform better when paired with a simpler ODE solver in practice? It's also a bit counter-intuitive when the function is stiff for higher-order solvers but not for simpler ones. Could the authors give some illustrative examples? The review has one plausible hypothesis: the gradient of classifier (condition function) has large variance or changes rapidly across different $\sigma$. Hence, it's harmful to combine the evaluations along the ODE trajectory.

- The evaluation protocol in Section 4.1 is a bit problematic. The generated samples are compared against DDIM w/ 1000 steps. It assumes that the samples generated by DDIM w/ 1000 steps are true samples. As we see in hindsight, there are many samplers better than DDIM, and they could generate better samples while incurring high LPIPS. Also, in this setup, the goal is to purely decrease the ODE simulation error (there is nothing to do with network estimation error). It's counter-intuitive that higher-order methods do not work out under such a metric.


### Writing suggestions

Below are some points that could potentially make the paper more readable and consistent:

- Section 3.1 & 3.2: the $\sigma$_s is increasing over time. But normally, they should decrease over time during sampling.
- Section 4, page 5, "with 1000 total steps" - do you mean "with 1000 total sampling steps"?
- Section 4.1: It would be helpful to change PLMS1 to Euler's method.





**Summary Of The Paper:**

The paper considers the problem of accelerating the sampling process of the classifier-guided diffusion models. The classifier-guided diffusion consists of two components: the conditional function (gradient of classifier) and the standard diffusion term. The authors observe that high-order numerical solvers can lead to degraded performance when sampling with the conditional function. To alleviate the issue while speed up the sampling, they opt to use higher-order method for standard diffusion term and forward Euler method for conditional function. They combine the two interleaved procedures through operator splitting methods. Experimentally, the proposed sampling method outperforms DDIM in speed and sample quality.

**Summary Of The Review:**

The reviewer leans toward rejection due to the lack of justification of the proposed method, and the seemingly low practical value of the considered (classifier-guidance) problem.

---

> ### Author Response · Authors · 2022-11-16
> **Answer to Reviewer PHsB (part 1/2)**
>
> We thank the reviewer for great suggestions and concerns. We believe those inquiries significantly improved our work.
>
> ### practical value of  classifier-guidance
> >The current dominating method is classifier-free generation, such as stable diffusion. The problem of classifier-guided diffusion models seems to have lower practical value.
> - We only use classifier-guided sampling as a demonstrative example. Our method can be applied to a wide range of diffusion problems, some mentioned in our paper, and some tested in our paper. (We suspected that you might use the term “classifier-guidance” to refer to the broader “gradient guidance” problem our paper mainly focuses on. “Classifier guidance” is a subset of “gradient guidance” or simply “guided diffusion” used in our paper) Importantly, gradient-guided and classifier-free guidance are *not* mutually exclusive. They are orthogonal and can be combined to improve the sampling outcome further. Our splitting method can be applied to any diffusion model that can be expressed as an ODE with a sum of two terms, which includes classifier-free diffusion models. After our submission, new findings from a new paper [2] (October 2022) and the Stable Diffusion community [3] indicate that using CLIP’s gradients to guide Stable Diffusion helps improve image quality. You can find our experiment comparing splitting methods on CLIP-guided Stable Diffusion in [4], which is now also a part of our Appendix M. In addition, we discovered that Dreambooth Stable Diffusion (fine-tuned Stable Diffusion) sometimes has trouble working with high-order methods, and the splitting methods can similarly help reutilize high-order methods. This experiment on classifier-free diffusion can be found in [5] and Appendix L. However, the issues of classifier-free diffusion models may differ from those of classifier-guided diffusion and require a future in-depth investigation.
> - Gradient-guided diffusion is still being actively investigated in recent work, such as [1] from ICASSP 2023, which solves three audio-related tasks: audio bandwidth extension, audio inpainting, and audio declipping. And the incorporation of classifier-guided in [2].
> - We believe that much of the potential of guided diffusion remains unexplored and ongoing, and the fact that pure classifier-free is the trend or more prevalent may even further warrant our contributions, which broaden the knowledge of this underexplored yet promising area---we do believe that striving for breadth of understanding is crucial, as a field. Our study of splitting numerical methods in guided-diffusion ODE is novel and unique in this regard as well as applicable and relevant to many recent studies released after our submission.
>
> - .[1]. (Moliner et al.) [CQTDiff: Solving audio inverse problems with a diffusion model]([https://arxiv.org/abs/2210.15228](https://arxiv.org/abs/2210.15228)) ICASSP 2023
> - .[2]. (Li et. at.) [UPainting: Unified Text-to-Image Diffusion Generation with Cross-modal Guidance](https://arxiv.org/abs/2210.16031) 2022
> - .[3]. [Diffuser Github](https://github.com/huggingface/diffusers/tree/main/examples/community#clip-guided-stable-diffusion)
> - .[4]. [Colab Notebook: CLIP-Guided Stable Diffusion](https://colab.research.google.com/drive/1uDArGUikVwuNVPX6KRVnSxIjfd6vJeZ1?usp=sharing)
> - .[5]. [Colab Notebook: Dreambooth Stable Diffusion](https://colab.research.google.com/drive/1xm3JZgh_DR6GJnlmmcz36SiLZ0ECVVqp?usp=sharing)
>
>
> ### compare classifier-free and classifier-guided on FID
> >Could the authors compare the FID/sampling time of the two methods, in order to demonstrate the utility of classifier-guided generation.
> - Thank you for your suggestion. Unfortunately, we could not complete this study given the limited rebuttal time. However, as previously stated, our contributions are not specific to classifier-guidance and classifier-free is not our baseline competitor. Though, some existing comparisons can be found in [1] and [2]. In general, classifier-guided and classifier-free guidance are orthogonal and can be combined to improve sampling quality further.
> -  .[1]  Ho and Salimans, [Classifier-free diffusion guidance](https://arxiv.org/abs/2207.12598), 2022
> -  .[2]  Nichol et al, [GLIDE: Towards photorealistic image generation and editing with text-guided diffusion models](https://arxiv.org/abs/2112.10741), 2021

---

> > ### Author Response · Authors · 2022-11-16
> > **Answer to Reviewer PHsB (part 2/2)**
> >
> > ### Illustrative examples
> > >why does the condition function perform better when paired with a simpler ODE solver in practice? It's also a bit counter-intuitive when the function is stiff for higher-order solvers but not for simpler ones. Could the authors give some illustrative examples?
> > - Thank you for your suggestion. Following your advice, we have created a toy example to illustrate how adding more terms to an ODE can cause failure / divergence to non-split high-order methods. The experiment can be found in [1], and the result of this experiment was added to Appendix O. This experiment shows how the non-splitting methods can fail even on a linear problem.
> > - We also added a stability analysis of different numerical methods on our toy problem in Appendix P. We also provided a study on the minimum number of steps before each method is theoretically guaranteed to diverge from the exact solution. The result shows that the non-splitting methods are more prone to failure than the splitting methods.
> > - .[1]. [Colab Notebook: Toy Example](https://colab.research.google.com/drive/1j6Vr-yuDXdlkmsq69pFfC29adHykfZX6?usp=sharing)
> >
> > ### Reviewer's  hypothesis
> > >The review has one plausible hypothesis: the gradient of classifier (condition function) has large variance or changes rapidly across different σ. Hence, it's harmful to combine the evaluations along the ODE trajectory.
> > - Thank you for this hypothesis. In the past, we also suspected that the variance of classifier gradients might be the root cause of our problem. However, our experiment showed that sometimes the variance of the diffusion term is higher than that of the classifier gradient and high-order methods are working properly with only the diffusion term (unconditional sampling). So, we could not conclude that the variance (the changes across different σ) causes the problem.
> >
> >
> > ### The evaluation protocol
> > >The generated samples are compared against DDIM w/ 1000 steps. It assumes that the samples generated by DDIM w/ 1000 steps are true samples. As we see in hindsight, there are many samplers better than DDIM
> > - We did not consider DDIM@1000 as a baseline we need to beat, but DDIM@1000 serves as a fine-step approximation to the exact solution of ODE, which allows us to evaluate which numerical methods can best approximate the ODE solution. For this goal, since every method has its own convergence theory to confirm that its solutions converge to the same exact solution as DDIM, it does not matter which method to use as long as the number of steps is sufficiently high. Using DDIM@1000 can reasonably satisfy this requirement, while other studied samplers or high-order methods, which were tested on much lower sampling steps than 1,000 in the literature, can diverge under guided-diffusion problems as demonstrated in our paper. So for this purpose, using the first-order method of DDIM but with a sufficiently high number of steps is a sensible choice. On the other hand, we agree that there exists methods that can give a better FID using fewer steps than 1000 steps of DDIM. However, this is not the goal of our study or the intended purpose of our DDIM@1000.
> >
> > >they could generate better samples while incurring high LPIPS.
> > - To ensure that method also generates better sample quality, we provide alternative evaluations in FID, improved Precision and Recall in Table 2, and newly added experiment FID vs. sampling time in Figure 9 Appendix I. The results show very similar conclusions to those shown by LPIPS.
> >
> > ### Writing errors
> > - Thank you. We have fixed them in our revision.
> >
> >
> > ======== additional answer =====
> > ### Minor change
> > > Section 3.1 & 3.2: the σ_s is increasing over time. But normally, they should decrease over time during sampling.
> > - Thank you for pointing this out. We fixed them and also added “Note that $\sigma_n$ is a decreasing sequence” to avoid future confusion.
> >
> > > Section 4, page 5, "with 1000 total steps" - do you mean "with 1000 total sampling steps"?
> > - Yes.  We have changed them to “with 1000 total sampling steps”  in our revision.
> >
> > > Section 4.1: It would be helpful to change PLMS1 to Euler's method.
> > - We have changed “PLMS1” to “Euler’s” in both text and figures.

---

> > ### Comment · Reviewer_PHsB · 2022-12-09
> > **Response / Some additional questions**
> >
> > I would like to thank the authors for demonstrating the utility of the proposed method on CLIP-guided algorithm and Dreambooth. Given the potential practical value, I raise the score to 6.
> >
> > However, there are still some unsettled questions. I appreciate the stability analysis in Appendix P, but the performance gain of STSP2 in this toy example seems a bit trivial: it essentially halved the step size for the fast-changing term (the term involves $s$) in Euler method, hence allowing for a lower number of steps $N$ in Table (10). In addition, the hand-crafted composition of a linear slow-changing and fast-changing term does not coincide with the diffusion models, per the authors' response ("we could not conclude that the (fast) changes across different $σ$ causes the problem."). The necessity of the splitting method in classifier-guidance is not fully understood in principle. Do the authors have any other thoughts on the problem?
> >
> >
> > Typo:
> >
> > Above Eq (51): (PLMS2) in Equation (44) -> (PLMS2) in Equation (49)

---

> > > ### Author Response · Authors · 2022-12-11
> > > **Respond to reviewer's additional questions**
> > >
> > > We first address some possible misunderstanding about the goal of our toy example and Appendix P:
> > > ## the purpose of our toy example
> > > - The goal of our toy example is to show how adding a new term can cause high-order numerical methods to fail and is for the original reviewer’s question:
> > > > why does the condition function perform better when paired with a simpler ODE solver in practice? It's also a bit counter-intuitive when the function is stiff for higher-order solvers but not for simpler ones.
> > >
> > > It was not used as evidence for our rebuttal statement:
> > > > we could not conclude that the (fast) changes across different σ  causes the problem.
> > >
> > > - To achieve that goal, we need an example ODE that yields an exact solution so that theoretical analyses can be made. Designing a toy ODE that also numerically “coincides” with diffusion models that use neural networks was never our goal, nor is it a tractable one for theoretical analyses. Nonetheless, our toy example illustrates one possible existence of this counterintuitive phenomenon.
> > >
> > > ## the trivialness of Appendix P
> > > - If we are only considering non-split numerical methods, like pure Euler or PLMS, halving the step size will always trivially lead to $N/2$ number of steps before divergence. However, for STSP that involves halving the Euler step of the condition term and performing a high-order method on the diffusion term, there are two opposing factors at work that affect $N$: 1. Halving the Euler step generally leads to a lower $N$, but 2. high-order numerical steps usually require a higher $N$. It is not obvious to us whether Factor 1 always prevails over Factor 2, and if so, by how much. Our analysis not only shows that the $N$s of the splitting methods are indeed lower but also allows the exact $N$ to be computed analytically for these methods.
> > >
> > >
> > > ## fast/slow-changing terms
> > > - We'd like to clarify a few points about our rebuttal response: "our experiment showed that sometimes the variance of the diffusion term is higher than that of the classifier gradient." The mentioned experiment is not the toy problem in the Appendix and is not in the main paper. In this experiment, we randomly choose a starting point from the Gaussian distribution and then measure the norm of change of $\epsilon(x)$ and $\nabla f(x)$ across time steps on a diffusion model trained on ImageNet (the same one used in Exp 4.1, 4.2). The result shows that $\epsilon(x)$ sometimes changes faster than $\nabla f(x)$ and sometimes slower. It is unclear from the result how fast the changes must be to cause problems or how slow to avert problems. As a result, we do not want to draw any conclusions about the cause of failure due to fast-slow changing terms, including the reviewer's hypothesis, due to the lack of evidence.
> > >
> > > - Similarly, there are no clear slow-changing and fast-changing terms in our toy example's problem. The change of $\epsilon(x)$ term is usually slower than that of $s\cdot g(x)$, but in some situations, such as when $s=3, \Delta t = 1/6,$ and $x(0) =[1, -0.95]$, the $\epsilon(x)$ term can change faster than $s\cdot g(x)$, and the ODE is still stiff (see [figure](https://ibb.co/C8p4V8x)).
> > >
> > > - Although we do not truly understand the impact of fast/slow-changing terms *in ODE problems*, we do know that slow-changing and fast-changing terms **in the exact solution** are strongly related to the stiffness of the ODE [1]. In our toy example, the term with $e^{-(s+1)t}$ in the exact solution decays much faster than the term with $e^{-t}$, which causes high-order numerical failures (see more detail on [1], [Wiki](https://en.wikipedia.org/wiki/Stiff_equation)). (Note that these terms are parts of the exact solution of the ODE not the ODE itself) Any varying behavior of the terms in the exact solution of guided-diffusion problems can potentially be a source of numerical failures; however, the exact solution of ODE modeled by neural networks can not be easily computed analytically.
> > >
> > > [1] Lambert, J. D. (1992), _Numerical Methods for Ordinary Differential Systems_
> > >
> > > ## necessity of splitting method in principle
> > > - Unfortunately, we cannot theoretically analyze much about the guided-diffusion problems because we do not know the behavior of the terms in their exact solution. Other theoretical foundations required for such an analysis are also currently lacking from the literature, such as the stiffness of ODE or analytical behavior of neural ODE. Nonetheless, we hope that our findings and analyses in this work will inspire future research both in theory and practice in this area.
> > >
> > > - To restate our key finding: in classifier guidance, we know that classical high-order numerical methods fail easily at low numbers of steps and low-order numerical methods have worse accuracy. Both of them still require a large computational time to achieve low errors. The splitting methods can approximate solutions better than low-order numerical methods and are more resistant to divergence than high-order numerical methods.

---

### Official Review · Reviewer_3tU3 · 2022-10-24

**Confidence:** 3
**Clarity, Quality, Novelty And Reproducibility:** The paper is well written.
**Correctness:** 3
**Technical Novelty And Significance:** 2
**Empirical Novelty And Significance:** 3
**Recommendation:** 6

**Strength And Weaknesses:**

Strength:

The paper proposes a simple method that accelerates the guided diffusion sampling. The proposed method is efficient and can be applied to other problems, e.g., super-resolution, colorization, for acceleration. Experimental analysis shows the effectiveness of the proposed method.

Weaknesses:

I mainly concern the limited theoretical analysis about the proposed method.

The paper claims that the high-order numerical methods are unsuitable for the conditional function. However, it does not explain this finding clearly. It would be better to provide a theoretical analysis of this finding.

The paper claims that the Strange splitting algorithm can be proved to have better accuracy. However, there is no theoretical analysis.

**Summary Of The Paper:**

This paper proposes an acceleration method of guided diffusion sampling based on splitting numerical methods. Based on the finding that the high-order numerical methods are unsuitable for the conditional function, it develops a method based on Strang splitting and a combination of fourth and first-order numerical methods. Experimental results show that the proposed can accelerate the guided diffusion sampling.

**Summary Of The Review:**

Although the experimental results are good, I expect the authors could provide more theoretical analysis about the proposed method.

---

> ### Author Response · Authors · 2022-11-16
> **Answer to Reviewer 3tU3**
>
> We thank the reviewer for the positive feedback.
> ### Theoretical analysis
> >I mainly concern the limited theoretical analysis about the proposed method.
> - We thank the reviewer for the suggestion. We added many theoretical analyses to our Appendix, which includes:
> 	- Appendix N contains a convergence analysis of Lie-Trotter and Strang Splitting methods. The analysis shows that both splitting methods converge to the solution when the step length is small and that the Strang splitting method has a better convergence order than Lie-trotter splitting.
> 	- Appendix P contains a stability analysis of splitting methods versus non-splitting methods. The analysis shows that the non-splitting methods can quickly diverge from exact solutions when the number of steps is reduced.
> 	- Appendix O contains a toy example that demonstrates how high-order numerical methods can become unstable and divergent on a certain class of ODE problems (Stiff equation). We also provide a colab for this experiment in [1]. The toy example makes similar conclusions as the real cases in our main paper. The splitting methods can avoid the solution-diverge problem and appropriately fit the exact solution better than the non-splitting methods.
> -  .[1]. [Colab Notebook: Toy Example](https://colab.research.google.com/drive/1j6Vr-yuDXdlkmsq69pFfC29adHykfZX6?usp=sharing)

---

> > ### Comment · Reviewer_3tU3 · 2022-12-09
> > **No further comments on this manuscript**
> >
> > Thanks for the clarification. My concerns have been solved.

---

### Official Review · Reviewer_sMbW · 2022-10-26

**Confidence:** 4
**Correctness:** 3
**Technical Novelty And Significance:** 3
**Empirical Novelty And Significance:** 2
**Recommendation:** 6

**Clarity, Quality, Novelty And Reproducibility:**

**Clarity:** The paper is easy to read and follow.

**Quality:** The paper is of mediocre quality. As discussed above, baseline comparisons are insufficient, some relevant work is not cited or discussed, and the mathematical analyses of the proposed integration scheme are lacking.

**Novelty:** The method is novel. It is a sensible approach for classifier-guided sampling, but the overall significance is somewhat limited, because the other important setting of classifier-free guidance is not covered.

**Reproducibility:** There are no concerns with respect to reproducibility. The submission also includes code.

(Small note: Eq. (4) has a typo. I believe it should be $e_4$ on the far right)

**Strength And Weaknesses:**

**Strengths:**

- The idea to split the diffusion model's main score and classifier term in the generative ODE into separate parts, and solving them separately, is novel and certainly seems like a good and sensible idea.

- The analyses that are performed in Figure 2 are interesting and insightful. They nicely show that the additional classifier term is the problem that makes higher-order methods break down in the classifier-guidance setting.

- The experimental results on classifier-guided diffusion model sampling support the value of the proposed method.

**Weaknesses:**

- Most state-of-the-art conditional diffusion models these days rely on classifier-free guidance for conditional sampling. Unfortunately, the proposed methodology only applies to classifier guidance. This reduces the significance of the proposed method.

- The baseline comparisons are insufficient. Comparisons to the recent state-of-the-art solvers DEIS [1] and DPM solver [2] are missing. Furthermore, for the experiments on inpainting, colorization and super-resolution, only qualitative comparisons are presented. Also, only DDIM is considered there for comparison.

- The paper lacks a proper theoretical analyses of the proposed methods. Convergence, order and stability of the Lie-Trotter Splitting and Strang Splitting techniques are nowhere discussed. It is insufficient to just write sentences like "This method can be proved to have better accuracy". Moreover, I believe the deep learning community is not deeply familiar with these splitting schemes, so a much more thorough discussion and analysis is needed.

- The statement that "no prior diffusion work uses splitting numerical methods" is incorrect. Critically-damped Langevin Diffusion [3] uses Strang Splitting to develop a sampler for their setting (see their Section 3.3). This work is not cited or discussed at all.


[1] Zhang and Chen, Fast Sampling of Diffusion Models with Exponential Integrator, 2022

[2] Lu et al., DPM-Solver: A Fast ODE Solver for Diffusion Probabilistic Model Sampling in Around 10 Steps, 2022

[3] Dockhorn et al., Score-Based Generative Modeling with Critically-Damped Langevin Diffusion, 2021

**Summary Of The Paper:**

This paper analyzes classifier-guided sampling of diffusion models and observes that applying higher-order methods for accelerated sampling from the model does not work well in the guidance scenario. The root cause for that is that the additional guidance term defined by the classifier makes the generative ODE harder to solve. Explicit higher-order integrators do not seem to be suitable for the resulting stiff classifier-guided ODE. Based on that observation, the work proposes to split the ODE into two separate terms, the diffusion model's score term and the classifier term. These are then solved separately with different integrators and afterwards combined, using numerical splitting methods. The paper shows improved performance of classifier-guided diffusion model sampling over some selected baselines when using a limited number of synthesis steps.

**Summary Of The Review:**

In summary, this paper proposes some interesting and sensible ideas to improve classifier-guided diffusion model sampling. However, the paper is lacking on multiple fronts (see weaknesses above). (a) Baseline comparisons are insufficient. (b) The proposed splitting methods lack theoretical discussion and are only very superficially discussed in the paper. (c) The significance is somewhat limited because only classifier guidance, but not classifier-free guidance is covered.

Therefore, overall I do not think that this paper is ready for publication in its current form. Most importantly, I would suggest the authors to include more thorough experiments and a more in-depth analysis and discussion of the proposed splitting integration methods.

---

> ### Author Response · Authors · 2022-11-16
> **Answer to Reviewer sMbW (part 1/2)**
>
> We thank the reviewer's thoughtful feedback and comments. We are delighted that the reviewer thinks our methods are sensible and novel. We believe the reviewer raised several helpful questions, and we believe those inquiries significantly improved our work.
>
> ### methodology only applies to classifier guidance
> > Unfortunately, the proposed methodology only applies to classifier guidance. This reduces the significance of the proposed method.
> - We thank the reviewer for the concern. First, our method is *not* limited to classifier guidance, but is applicable to *gradient guidance* and *classifier-free* models, or more broadly any method where the diffusion ODE can be formulated as a summation of two terms. We did show three gradient-guidance applications in our original submission in Section 4.3, and we also show a successful acceleration of a classifier-free model, i.e., a fine-tuned Stable Diffusion model, Dreambooth, in our new Appendix L.
> (We suspected that you might use the term “classifier-guidance” to refer to the broader “gradient guidance” problem our paper mainly focuses on. “Classifier guidance” is a subset of “gradient guidance” or simply “guided diffusion” used in our paper. Nonetheless, our method also works on classifier-free models (Appendix L) but they are orthogonal to our contribution.)
>
>
> ### diffusion models these days rely on classifier-free guidance
> > Most state-of-the-art conditional diffusion models these days rely on classifier-free guidance for conditional sampling.
>
> > but the overall significance is somewhat limited, because the other important setting of classifier-free guidance is not covered.
>
> - Gradient-guided and classifier-free guidance are *not* mutually exclusive. They are orthogonal and can be combined to improve the sampling outcome further. Our splitting method can be applied to any diffusion model that can be expressed as an ODE with a sum of two terms, which includes classifier-free diffusion models. After our submission, new findings from a new paper [2] (October 2022) and the Stable Diffusion community [3] indicate that using CLIP’s gradients to guide Stable Diffusion helps improve image quality. You can find our experiment comparing splitting methods on CLIP-guided Stable Diffusion in [4], which is now also a part of our Appendix M. In addition, we discovered that Dreambooth Stable Diffusion (fine-tuned Stable Diffusion) sometimes has trouble working with high-order methods, and the splitting methods can similarly help reutilize high-order methods. This experiment on classifier-free diffusion can be found in [5] and Appendix L. However, the issues of classifier-free diffusion models may differ from those of classifier-guided diffusion and require a future in-depth investigation.
> - Gradient-guided diffusion is still being actively investigated in recent work, such as [1] from ICASSP 2023, which solves three audio-related tasks: audio bandwidth extension, audio inpainting, and audio declipping. And the incorporation of classifier-guided in [2].
> - We believe that much of the potential of guided diffusion remains unexplored and ongoing, and the fact that pure classifier-free is the trend or more prevalent may even further warrant our contributions, which broaden the knowledge of this underexplored yet promising area---we do believe that striving for breadth of understanding is crucial, as a field. Our study of splitting numerical methods in guided-diffusion ODE is novel and unique in this regard as well as applicable and relevant to many recent studies released after our submission.
> - .[1]. (Moliner et al.) [CQTDiff: Solving audio inverse problems with a diffusion model]([https://arxiv.org/abs/2210.15228](https://arxiv.org/abs/2210.15228)) ICASSP 2023
> - .[2]. (Li et. at.) [UPainting: Unified Text-to-Image Diffusion Generation with Cross-modal Guidance](https://arxiv.org/abs/2210.16031) 2022
> - .[3]. [Diffuser Github](https://github.com/huggingface/diffusers/tree/main/examples/community#clip-guided-stable-diffusion)
> - .[4]. [Colab Notebook: CLIP-Guided Stable Diffusion](https://colab.research.google.com/drive/1uDArGUikVwuNVPX6KRVnSxIjfd6vJeZ1?usp=sharing)
> - .[5]. [Colab Notebook: Dreambooth Stable Diffusion](https://colab.research.google.com/drive/1xm3JZgh_DR6GJnlmmcz36SiLZ0ECVVqp?usp=sharing)

---

> > ### Author Response · Authors · 2022-11-16
> > **Answer to Reviewer sMbW (part 2/2)**
> >
> > ### Comparisons to DEIS and DPM-solver are missing
> > - We did compare the core numerical methods of both methods in our paper. The method in DMP-solver is RK2 which belongs to the RK’s family. We pick RK2 and RK4 as representatives in Figure 3. The methods in DEIS belong to the linear multistep family, which we pick PLMS as representative in Figure 3. Due to slight differences between DEIS and PLMS, we derive another linear multistep method based on DEIS in our Appendix B. We demonstrate that it performs similarly to PLMS and that DEIS will converge to PLMS if the number of steps is high.
> > - We added a more thorough comparison of the similarities between DEIS and PLMS in our Appendix D.
> > - We added comparisons between our splitting methods and both DEIS and DPM-Solver. This comparison is made by implementing our splitting method into both papers' original implementations. The results have been added to Tables 7 and 8 in Appendix H.
> >
> > |                    | 5 sec     | 10 sec    | 15 sec    | 20 sec    |
> > | ------------------ | --------- | --------- | --------- | --------- |
> > | DPM-solver-1(DDIM) | 0.333     | 0.125     | 0.080     | 0.045     |
> > | DPM-Solver-2       | 0.565     | 0.188     | 0.078     | 0.045     |
> > | DPM-Solver-3       | 0.540     | 0.233     | 0.087     | 0.043     |
> > | **LTSP4**          | 0.185     | 0.105     | 0.071     | 0.048     |
> > | **STSP4**          | **0.169** | **0.062** | **0.061** | **0.037** |
> >
> > |               | 3 sec.    | 6 sec.    | 9 sec.    | 12 sec.   |
> > | ------------- | --------- | --------- | --------- | --------- |
> > | 0-DEIS (DDIM) | 0.333     | 0.125     | 0.080     | 0.045     |
> > | 1-DEIS        | 0.466     | 0.193     | 0.092     | 0.044     |
> > | 3-DEIS        | 0.625     | 0.511     | 0.433     | 0.345     |
> > | **LTSP4**     | 0.321     | 0.120     | 0.080     | 0.048     |
> > | **STSP4**     | **0.212** | **0.080** | **0.046** | **0.031** |
> >
> > ### only qualitative comparisons are presented. Also, only DDIM is considered
> > > for the experiments on inpainting, colorization and super-resolution, only qualitative comparisons are presented. Also, only DDIM is considered there for comparison.
> > - Our original submission did have quantitative results and comparisons to other methods, such as PLMS4 and LTSP4, in Appendix K.
> >
> > ### Convergence, order and stability are nowhere discussed.
> > > Convergence, order and stability of the Lie-Trotter Splitting and Strang Splitting techniques are nowhere discussed
> > - Thank you for your advice. A convergence analysis of Lie-Trotter Splitting and Strang Splitting has been added to Appendix N, and a stability analysis that supports the stability of splitting methods over non-splitting methods has been added to Appendix P. More citations are also included to corroborate our claims.
> >
> > ### Related Work
> > >The statement that "no prior diffusion work uses splitting numerical methods" is incorrect. Critically-damped Langevin Diffusion [3] uses Strang Splitting to develop a sampler for their setting (see their Section 3.3).
> > - Thank you. We revised the statement from "no prior diffusion work uses splitting numerical methods" to "no prior guided-diffusion work uses splitting numerical methods" as that paper applies Strang Splitting on an SDE, not *guided-*diffusion ODE. We added a citation to this paper in our revision.

---

> > > ### Comment · Reviewer_sMbW · 2022-11-22
> > > **Thank you for the response**
> > >
> > > I would like to thank the authors for their reply, for answering my questions and for adding many additional results to the paper. I think some of my major concerns could be addressed. In particular, I can now see how the method can also be used in the context of classifier-free guidance. Moreover, the examples on additional types of guidance (CLIP, etc.) are interesting. I also appreciate the additional comparisons to DEIS and DPM-Solver. I do not have any further questions.
> > >
> > > Overall, I significantly raised my score and am now leaning towards suggesting acceptance.

---

### Official Review · Reviewer_CQRF · 2022-10-27

**Confidence:** 3
**Correctness:** 3
**Technical Novelty And Significance:** 3
**Empirical Novelty And Significance:** 2
**Recommendation:** 6

**Clarity, Quality, Novelty And Reproducibility:**

Overall, the paper is easy to follow.  The newly designed sampling methods LTSP and STSP sound reasonable, where the ODE term from the classifier contribution is solved by a first order method and the ODE term from the diffusion contribution is solved by a high-order method.

There are a few inconsistent statements:
1. On page 4 of experiments, it says DDIM at 1000 steps are taken as reference solutions  while in Figure 2 and 3 later on, DDIM of 250 steps are considered.
2. On page 6, it says condition subproblem (Equation 10)... diffusion subproblem (Equation 11), which I believe is a mistake.

I think it is more convincing to replace LPIPS in Figure 2 and 3 with FID, which most readers are more familiar with.

**Strength And Weaknesses:**

Strength:
1. The paper, for the first time, applies existing splitting methods (LTSP 1959 and Strange splitting 1968) for solving the backward ODE in guided diffusion sampling and obtain promising sampling results, which I think is interesting and novel.
2. The literature over existing high order methods seems up-to-date.

Weaknesses:
It is unclear from a theoretical point of view why non-splitting high-order methods do not work well. There may exist other high-order methods that work which yet to be discovered.






**Summary Of The Paper:**

This paper focuses on accelerating guided diffusion sampling, where the backward ODE to be solved consists of two parts: the first one from the diffusion contribution and the second one from the classifier contribution. It is empirically found that non-splitting high-order ODE solvers does not work well when the number of time steps is small. To address the above issue, the authors consider applying splitting methods (LTSP 1959 and Strange splitting 1968) for solving the backward ODE, where the two parts in the ODE are treated separately and sequentially.

**Summary Of The Review:**

The paper tried out splitting methods for solving the backward ODE in guided diffusion sampling and obtained better performance. No theoretical justification is provided to motivate the splitting methods over non-splitting methods.

---

> ### Author Response · Authors · 2022-11-16
> **Answer to Reviewer CQRF**
>
> We thank the reviewer for your time and useful suggestions to help strengthen the paper.
> ### DDIM at 1000 step and 250 steps
> > On page 4 of experiments, it says DDIM at 1000 steps are taken as reference solutions while in Figure 2 and 3 later on, DDIM of 250 steps are considered.
> - DDIM 1000 and 250 are used for different purposes. DDIM@1000 represents a fine-step approximation to the exact solution of ODE, while DDIM@250 is a baseline commonly used in many papers because it produces visually reasonable results. Figure 2 and 3 also need DDIM@1000 to provide a reference exact ODE solution for computing each method’s LPIPS score.
>
> ### Readers are more familiar with FID score
> > I think it is more convincing to replace LPIPS in Figure 2 and 3 with FID, which most readers are more familiar with.
> - We evaluate FID vs. sampling time on DDIM, PLMS4, RK2, RK4, LTSP4 and STSP4 . The additional results are reported in Figure 9, Appendix I. We discover that non-splitting numerical methods converge slower than splitting numerical methods. They require a longer sampling time to achieve the same level of FID as the splitting methods.
> - [Figure: FID vs time]([https://i.ibb.co/Wzg1QCw/FID.png](https://i.ibb.co/Wzg1QCw/FID.png))
>
> ### The splitting method theory
> > No theoretical justification is provided to motivate the splitting methods over non-splitting methods
> -  We added many theoretical analyses to our Appendix, which includes:
> 	- Appendix N contains a convergence analysis of Lie-Trotter and Strang Splitting methods. The analysis shows that both splitting methods converge to the solution when the step length is small and that the Strang splitting method has a better convergence order than Lie-trotter splitting.
> 	- Appendix P contains a stability analysis of splitting methods versus non-splitting methods. The analysis shows that the non-splitting methods can quickly diverge from exact solutions when the number of steps is reduced.
> 	- Appendix O contains a toy example that demonstrates how high-order numerical methods can become unstable and divergent on a certain class of ODE problems (Stiff equation). We also provide a colab for this experiment in [1]. The toy example makes similar conclusions as the real cases in our main paper. The splitting methods can avoid the solution-diverge problem and appropriately fit the exact solution better than the non-splitting methods.
> -  .[1]. [Colab Notebook: Toy Example]([https://colab.research.google.com/drive/1j6Vr-yuDXdlkmsq69pFfC29adHykfZX6?usp=sharing](https://colab.research.google.com/drive/1j6Vr-yuDXdlkmsq69pFfC29adHykfZX6?usp=sharing))
>
> ### Wording
> >There may exist other high-order methods that work which are yet to be discovered.
> - Thank you. This is correct and we have revised our wording from  "high-order methods" to "classical high-order methods".
>
> ### Writing errors
> - Thank you. We have fixed them in our revision.

---

### Author Response · Authors · 2022-11-17
**A Summary of Paper Updates**

We would like to thank all of the reviewers for their insightful comments. We addressed all of the comments, questions, and concerns, and we believe it has improved the paper significantly. This has strengthened the paper's findings and claims. We have updated our main paper pdf and supplementary material.

Following their suggestions, we corrected the typos and made the following major changes to the paper, including the addition of *8 new sections* in our Appendix to provide more justifications, theories, and examples for our paper:
1. **Appendix H**: add more comparisons between PLMS and DEIS
2. **Appendix F**: add a p-value table to support the main experiment in our paper
3. **Appendix I**: an additional experiment on FID vs. sampling time
4. **Appendix H**: an additional experiment to compare our method with DEIS and DPM-solver.
5. **Appendix L**: an example of using our method on Dreambooth Stable Diffusion (Classifier-free diffusion model)
6. **Appendix M**: an example of using our method on CLIP-Guided Stable Diffusion (Gradient guide classifier-free diffusion model)
7. **Appendix N**: Convergence analysis of splitting methods
8. **Appendix O**: a toy example to illustrate the phenomenon in our main paper and how the splitting methods can help.
9. **Appendix P**: a stability analysis comparing splitting and non-splitting methods.

---

### Decision · Program_Chairs · 2023-01-20

**Decision:**

Accept: poster

**Justification For Why Not Higher Score:**

The presentation of the paper could be further improved. The experimental validation can be strengthened.

**Justification For Why Not Lower Score:**

The split numerical method is new to the diffusion model community and seems to be useful. We would like to make sure researchers in the field know such tools to help them conduct better reserch.

**Metareview: Summary, Strengths And Weaknesses:**

The paper reports that high-order ODE solvers for diffusion models do not work well when classifier guidance is used. It argues that this is due to the gradients from the score models and those from the classifiers having very different perspectives. To address the issue, the paper proposes to use the split numerical methods in the ODE literature to solve the issue. Experiment results on limited scenarios verify the effectiveness of the proposed method.

The paper receives 4 reviews. Initially, three reviewers considered the paper slightly above the bar, and one reviewer considered it slightly below the bar. Generally, the reviewers feel the proposed approach makes sense, but several were not convinced that classifier guidance is important as classifier-free guidance is now the dominant approach. The authors rebut the idea with evidence that classifier guidance remains relevant for the diffusion model community. In the end, the authors successfully convince the negative reviewer and reach a consensus above the bar evaluation.

After reading through the paper, reviews, and rebuttal, the area chair agrees with the reviewer's assessment. The paper does have merits to be presented at the ICLR conference despite the fact that the presentation was a bit below the bar. The authors are encouraged to incorporate the review feedback and strength the paper for the final version.

**Note From Pc:**

if the above contains the word "oral" or "spotlight" please see: "oral" presentation means -> notable-top-5% and "spotlight" means -> notable-top-25%. As stated in our emails, we are disassociating presentation type from AC recommendations

**Summary Of Ac-Reviewer Meeting:**

We scheduled a review meeting to discuss the paper. However, right before the meeting, all the reviewers converge to the acceptance rating and hence we did not discuss much in the meeting.